# Recycling Polyethylene/Polyamide Multilayer Films with Poly(isoprene-*g*-Maleic Anhydride) Compatibilizer

**DOI:** 10.3390/polym16081079

**Published:** 2024-04-12

**Authors:** Andreia Romeiro, Cidália Teixeira, Henrique Costa, Jorge F. J. Coelho, Arménio C. Serra

**Affiliations:** 1University of Coimbra, Centre for Mechanical Engineering Materials and Processes, ARISE, Department of Chemical Engineering, Rua Silvio Lima- Polo II, 3030-790 Coimbra, Portugal; andreiar@eq.uc.pt (A.R.); jcoelho@eq.uc.pt (J.F.J.C.); 2Componit, lda, Estrada Nacional 3 km 28.6, Vila Chã de Ourique, 2071-621 Santarém, Portugal; cidalia.paula@isolago.com; 3Inventive Matl, Casal dos Eucaliptos, Casais Lagartos, 2070-389 Pontével, Portugal; incentivos@inventivemat.pt

**Keywords:** polymer recycling, polyethylene, polyamide, polyisoprene–graft–maleic anhydride, compatibilization, monolayer film

## Abstract

Polymers generally form incompatible mixtures that make the process of recycling difficult, especially the mechanical recycling of mixed plastic waste. One of the most commonly used films in the packaging industry is multilayer films, mainly composed of polyethylene (PE) and polyamide (PA). Recycling these materials with such different molecular structures requires the use of compatibilizers to minimize phase separation and obtain more useful recycled materials. In this work, commercial polyisoprene–graft–maleic anhydride (PI-*g*-MA) was tested as a compatibilizer for a blend of PE and PA derived from the mechanical recycling of PE/PA multilayer films. Different amounts of PI-*g*-MA were tested, and the films made with 1.5% PI-*g*-MA showed the best results in terms of mechanical properties and dart impact. The films were also characterized thermally via thermogravimetric analysis (TG) and differential scanning calorimetry (DSC), using Fourier-transform infrared spectroscopy (FTIR), and morphologically using a scanning electron microscope (SEM). Other parameters, such as tearing and perforation, were analyzed.

## 1. Introduction

Packaging is an inevitability of commercial activity and the global transaction of goods. Hence, recycling plastic packaging is becoming increasingly important in preserving limited resources and protecting the environment from plastic waste [1]. In the case of food packaging, multilayer films are used to maintain the integrity of food and protect it against degradation processes. The justification for the different layers films is due to the different functions carried out by the different polymer films [2]. For example, low-density polyethylene (PE) or ethylene vinyl acetate copolymer (EVA) is commonly used to provide low-temperature heat seal ability, polyethylene terephthalate as the outside layer to improve printability and abrasion resistance, and polyamide 6 (PA) or poly(vinyl alcohol-co-ethylene) as a gas barrier layer [3]. So far, multilayer packaging is mostly incinerated or landfilled because it is very hard to recycle a material that can be used for the same function [4], and the functionality of the materials is greatly lost mainly due to the general incompatibility between polymers. The European Union (EU) strategy stipulates that by 2030, all plastic packaging placed on the EU market must be reusable or economically (mechanically) recyclable.

The ideal recycling process corresponds to separating the different components of multilayer films and reprocessing each layer separately. The mechanical recycling of multilayer films is an easy and attractive operation due to its simplicity and no need for specialized equipment. However, it is a big challenge since the plastic layers are generally not compatible with each other. Consequently, the mechanical recycling of multilayer films results in a single material with distinguishable phases and domains, poor mechanical properties, and barrier effects far below those of the components [3,4,5]. The simpler way to improve the quality of this material is to use compatibilizers to reduce phase segregation in the final product. Compatibility is usually promoted by adding block or graft copolymers with structural affinity for the different polymers to reduce the interfacial tension between the phases, improve dispersion, and promote adhesion [3,4,5,6]. 

PE/PA blends require a compatibilizing agent to achieve good interfacial adhesion between phases and improve mechanical, thermal, and permeability properties [7]. Some compatibilizers that have been used for low-density polyethylene/polyamide (LDPE/PA) blends are ethylene–acrylic acid copolymer [8,9,10], LDPE functionalized with acrylic acid [11], ethylene glycidyl methacrylate [12], ethylene vinyl acetate (EVA) [13], ethylene-methacrylic acid copolymer partially neutralized with sodium (Na-EMAA) [14], and maleic anhydride (MA) [1,15,16,17,18,19,20,21,22,23,24]. 

Much work has been performed to improve the compatibility of PE/PA blends using maleic anhydride groups. Moreno et al. [1] evaluated the effect of the concentration of PE- g-MA as a compatibilizer with blends of LDPE/PA6, confirming that the compatibilizer causes changes in the morphology and rheology of the blend and improves the mechanical performance. Anjos et al. [20] evaluate the effect of maleic anhydride-grafted linear low-density polyethylene (LLDPE-*g*-MA) as a compatibilizer on the rheological, thermal, mechanical, and morphological properties of PA6/LLDPE blends and find that the addition of the compatibilizer agent increases the impact strength of the 50/50 (PA6/LLDPE) blend by 160%. Moreno et al. [21] studied the properties of blends of LDPE and PA6 generated from mechanical recycling of multilayer films, using polyethylene–graft–maleic–anhydride (PE-*g*-MA) as compatibilizer and different amounts of virgin PA6. The blends were immiscible at all compositions, and with a high content of PA6, a minor effect of the compatibilizer was observed. Czarnecka-Komorowsk et al. [23] investigated the influence of maleic anhydride grafted polyethylene (PE-*g*-MAH) on the morphology and mechanical properties of the recycled PE/PA blends. It was found that the addition of PE-*g*-MAH was beneficial to the blend, and it was proved that it was possible to produce granules for industrial applications. Silva et al. [24] studied the effects of the poly(ethylene-alt-maleic anhydride) (HDPE-*alt*-MAH) compatibilizer on the mechanical properties of HDPE/PA12 blends. It was found that the addition of 2 wt% of compatibilizer was sufficient to produce blends with significantly better mechanical properties. 

There are publications where some polybutadienes are used in polyethylene mixtures [25] or modified polyamide properties [26,27,28,29,30]. Like natural rubber, polyisoprene compounds exhibit good building tack, high tensile strength, good hysteresis, and good hot tensile and hot tear strength. The presence of an anhydride group in polyisoprene could originate a compatibilizer for the PE/PA blend. In this work, we developed a process for recycling films of PE/PA made up of seven layers into a single material with improved properties using commercial polyisoprene–graft–maleic anhydride as a compatibilizer. The PE and PA constitute 70% of the total amount of polymers; others include poly(vinyl alcohol) (PVOH) and adhesive layers. The introduction of small amounts of polyisoprene-*g*-maleic anhydride improves the final film performance, particularly in terms of mechanical properties.

## 2. Materials and Methods

### 2.1. Materials

The material used in this study ((PE/PA)rec) is the result of the extrusion of a multilayer film composed of PE/PA (70%) and PVOH/adhesives (30%) obtained from industrial waste (courtesy of Danipack Lda, Estarreja, Portugal) In the mixture PE/PA the amount of PE is 80%. Polyisoprene–graft–maleic anhydride (PI-*g*-MA), average Mw ~25,000, was purchased from Aldrich (Sintra, Portugal). Amorphous precipitated silica, Tixosil^®^ 38, was provided by Componit Lda (Vila Chã de Ourique, Portugal).

### 2.2. Extruders

The (PE/PA)rec pellets were mixed with the polyisoprene–graft–maleic anhydride compatibilizer (1.5–6%) and 3% of tixosil to improve the mixing quality and extruded. The production of pellets was carried out in a double screw extruder in which the mixture of raw materials was dosed in a chamber with two screws. 

Table 1 and Table 2 show the temperature profile and the extrusion conditions to obtain the pellets.

### 2.3. Production of the Films

The next step was extruding the above pellets in a balloon extruder to obtain the films. Table 3 shows the equipment parameters for producing the films via a balloon extruder.

The obtained films have 100 µm of thickness.

### 2.4. Preparation of the Samples for Characterization

Solvent fractionation experiments were performed to investigate the composition of the binary PE/PA blends. The treatment of small samples with 85% formic acid (a solvent of the PA phase) was carried out at room temperature for 62 h. After this procedure, the samples were washed with ethanol and dried in an oven at 40 °C. 

### 2.5. Characterization Techniques

Thermogravimetric analysis (TGA) and Differential scanning calorimetry (DSC)

The thermal stability of films was studied using thermogravimetric analysis (TGA) that was conducted using NETZSCH TG 209F1 (Netzsch, Germany). Samples were heated in a temperature range of 30–600 °C at a heating rate of 10 K·min^−1^ under nitrogen purge flow. Also, thermal behavior was evaluated using differential scanning calorimetry (DSC) made in a NETZSCH DSC 204 F1 Phoenix model (Netzsch, Germany). All samples were analyzed in an aluminum pan with an ordinarily closed aluminum lid. The samples were heated from room temperature to 300 °C, then cooled to −50 °C, and followed a heating cycle to 300 °C. A heating/cooling/heating rate of 10 °C·min^−1^ was used. A dry nitrogen environment with a purge flow was applied. 

Fourier-transform infrared spectroscopy (FTIR)

Films were characterized using FTIR in ATR mode using an Agilent Technologies Carey 630 spectrometer equipped with a Golden Gate Single Reflection Diamond ATR in the 4000–600 cm^−1^ range at room temperature. Spectra were collected with 4 cm^−1^ spectral resolution and 64 scans. OMNIC software(version 8.2.0.387) was used to analyze spectra.

Scanning electron microscopy

To investigate the compatibilization of the polymeric blend, the fracture surfaces were analyzed via scanning electron microscopy (SEM). The specimens were frozen in liquid nitrogen prior to fracture to diminish the risk of plastic deformation. The fracture surfaces were coated with gold and analyzed with 1 kV of acceleration voltage in a field emission scanning electron microscope (FESEM), ZEISS MERLIN Compact/VPCompact, Gemini II. 

Tensile testing

Tensile tests were performed on an INSPEKT solo 2.5 mechanical tester equipped with a 500 N load cell. The film rectangular-shaped specimens were presented to tension at a rate of 50 mm·min^−1^ until failure. The thickness of films was measured with a digital micrometer screw gauge (precision 1 µm), and measurement was taken at three different locations on each film, and the mean value was used in the calculus of the mechanical test results. The present values are an average of five valid tests.

Dart Impact test and Tear test

Dart impact test (dart drop test) and tear test are often used when having blown and cast films manufactured from LDPE. The analyses of dart impact test were performed on CAST and Tear test on an Instron mechanical tester equipped with a 1 kN load cell.

## 3. Results and Discussion 

### 3.1. Film Preparation 

The films of (PE/PA)rec with the polyisoprene–graft–maleic anhydride compatibilizer were produced using tubular/balloon extrusion. As polyisoprene–graft–maleic anhydride is a viscous compound, to facilitate the processing, it was necessary to add a small amount of commercial silica (3%), since without the introduction of silica, the (PE/PA)rec pellets stick to the extruder’s feeding chamber, making it impossible to obtain homogeneous material. The silica impregnated with the desired amount of PI-*g*-MA disperses much better in the (PE/PA)rec pellets, making the final film more homogeneous. The small amount of silica also has the advantage of being a drying agent. 

### 3.2. Thermogravimetric Analysis 

Figure 1 shows the TGA thermograms of the samples. Figure 1a exhibits the graph of weight vs. temperature, and Figure 1b shows the weight derivate vs. temperature of obtained films with different amounts of PI-*g*-MA. 

The profiles of the different samples are very similar, indicating that the presence of the compatibilizer does not change the thermal stability of the blends (Table 4). The profiles present two major mass losses, one between 360 and 400 °C, with losses between 15 and 17%, and the second between 440 and 500 °C, with losses between 70 and 80%, where most of the material decomposes. The first one is probably due to the small amount of PVOH in the original multilayer film [31]. The second one corresponds to the decomposition of the PE/PA blend, which happens as a single event due to the proximity of the temperature decomposition of these polymers [7,20,32]. These events are much clearer on the DTGA curve, where a shoulder at lower temperatures precedes the main degradation step. The presence of the compatibilizer does not bring any significant changes in the degradation profile of the mixtures, except for a small shift in the maximum degradation temperatures. 

### 3.3. Differential Scanning Calorimetry 

DSC analysis was performed to evaluate the changes in thermal behavior caused by the introduction of the PI-*g*-MA compatibilizer, Figure 2, Figure 3 and Appendix A and Table 5 and Appendix A.

As can be seen in Figure 2 and Figure 3, all blends exhibit two zones characteristic of endothermic melting processes and two crystallization zones corresponding to the presence of immiscible polymers. The first zone between 109 °C and 120 °C corresponds to the polyethylene melting with broad signals and the melting of polyamide-6 as a single signal around 220 °C. The crystallization zones are between 50 °C and 115 °C for polyethylene and 160 °C to 190 °C for PA. The presence of a broad melting zone for polyethylene indicates the presence of different low-density polyethylenes on the recycled material derived from wasted multilayer films. Compared with the literature, the broad melting signal for LLDPE corresponds to the presence of crystals with different lamella thickness due to the heterogeneous distribution of short chain branches, which is a characteristic of LLDPE prepared using a Ziegler–Natta type catalysis. The peak corresponds to the crystallization of longer linear chains with less short-chain branch content, and the shoulders correspond to the crystallization of shorter linear chains with higher short-chain branch content [33,34,35]. Thermal events for the first heating are presented in Appendix A and Appendix A. Crystallization events (Figure 2 and Table 2) also show two major zones corresponding to the crystallization of the two polymers on the blend. A small crystallization event occurs at 62 °C for all the samples. This event is not currently seen [23] and is related to the nature of the LDPE. This event is explained by the melt topology and entanglements of PE chains during the crystallization process [35].

With the introduction of the compatibilizers small changes are observed in the DSC profiles for PE melting and crystallization. In relation to PA, the Tm is similar for all blends, but in relation to Tc, there are some differences. The addition of compatibilizers reduces the temperature of crystallization. This effect in PA is reported as a signal for the interaction between PA and the compatibilizer [36,37]. The effect of compatibilization via PI-*g*-MA can be better seen by analyzing the ∆H values for melting and crystallization of the blends [12]. For PE and PA, the ∆Hm value generally decreases only slightly compared to the ∆Hc value, which shows greater differences between the two polymers in the presence of the compatibilizer. With respect to PE, the compatibilizer increases the ∆Hc [38], and in the case of PA, the presence of compatibilizer decreases the ∆Hc, as observed by others [8,39]. This decrease in ∆Hc could be explained by fractional crystallization, in which the compatibilizer influences the size of the PA particles, causing small particles to crystallize at lower temperatures [8,39]. 

### 3.4. Infrared Spectroscopy (FTIR)

The ATR-FTIR spectra of the pellets of (PA/PE)rec blended with different amounts of PI-*g*-MA are shown in Figure 4.

The PE part is identified by the stretching vibrations of the -CH_2_ groups at 2912 cm^−1^ (asymmetric stretching) and 2842 cm^−1^ (symmetric stretching). The band at 1462 cm^−1^ is identified as C-H bending deformation [40,41]. The absorption bands at 728 cm^−1^ and 717 cm^−1^ correspond to the rocking deformations of the CH_2_ group [40]. 

The PA present in the blend shows the band corresponding to the amide group (N–H) at 3292 cm^−1^, while the C–H stretching bands of the methylene segments were observed at 2912 cm^−1^ and 2842 cm^−1^ [42,43]. The adsorption bands at 1636 cm^−1^ and 1557 cm^−1^ are associated with amide groups, the first with amide-I and the second with the amide-II forms [44]. The other band at 1462 cm^−1^ could be assigned to the CH_2_ bending of the amide group [41,42,43,45]. 

In the FTIR spectrum of PI-*g*-MA, the two absorption bands at 1791 cm^−1^ and 1712 cm^−1^ can be attributed to the anhydride ring, and the one weak absorption peak at 1656 cm^−1^ can be attributed to the C=C stretching vibration of polyisoprene [46]. 

The absence of bands corresponding to the carboxylic anhydride (two signals were assigned at 1856 cm−1 and 1780 cm−1) of the maleic anhydride ring (especially in the sample with 6% PI-*g*-MA) could suggest that the cyclic anhydride was reacted during processing.

To verify whether maleic anhydride reacted during processing, the FTIR spectrum of the sample in the form of pellets with 6% PI-*g*-MA (one processing) was compared with the spectrum of the same material in the form of film (two processings), Figure 5. Analysis of the spectra shows that, when the material is in pellets, there are small bands at 1705 cm^−1^ and 1739 cm^−1^ corresponding to the maleic anhydride band. These bands are no longer noticeable after the second processing of the material into a film. 

### 3.5. SEM

Figure 6 shows the SEM images of sections of cryo-fractured samples of the films made from de (PE/PA)rec with different amounts of PI-*g*-MA before and after etching the samples with formic acid to remove the PA fraction. Considering the incompatibility between PE and PA, it was expected that the morphology of the samples would show the typical distribution of one continuous phase of PE with PA droplets of different sizes across the continuous phase, as observed by others [7,37]. In our case, this pattern is not seen. The PE/PA sample shows the material in an elongated shape, like fibers that are not connected to each other. This different morphology could be because the films are subjected to stretching when blowing, and the materials, therefore, feel stretching forces and take on this shape [47]. In other works, the samples commonly resulted from injection or pressing processes, which implies compression forces and, therefore, a different morphology. The presence of the compatibilizer changes the structure into a lamellar and more compact one. With the removal of the PA part with formic acid (etched samples), a smoother surface appears, and a series of small craters are visible, corresponding to the PA phase dispersed in the PE. It is possible that some dissolved PA in the formic acid covers the surface, contributing to a more homogeneous surface.

### 3.6. Mechanical Performance

The mechanical performance of the samples was evaluated using the tensile–strain tests. The results are presented in Figure 7. The results show that, in general, when adding PI-*g*-MA, there is an increase in elongation at break compared with the sample without the PI-*g*-MA. Also, in relation to tensile strength, there is a small gain in the compatibilized samples, with 1.5% and 3% compatibilizers. These facts are indicative of the compatibilizing effect of the PI-*g*-MA polymer. The material with the highest elongation at break and the highest tensile strength is the sample with 1.5% PI-*g*-MA with gains relative to the 0% PI-*g*-MA of 16% and 44%, respectively. The presence of the reactive anhydride groups reacting with the PA portion and the isoprene part interacting with the PE part contributes to this improvement, probably due to the creation of positive interfacial interactions between PE and the end groups of polyamide-6. Similar results were presented by other authors [48,49], who noted that incorporating HDPE-*g*-MAH into blends improved hardness as a result of increased interfacial adhesion of phases in the blends. 

### 3.7. Tear Resistance, Perforation, and Impact Fall Dart 

In the industrial environment, important properties related to film applications were studied. Table 6 shows the results of the tearing, drilling, and dart drop tests. The tear resistance measures the resistance of the film to a tear propagation event. Films containing the compatibilizer (1.5 and 6%) show better properties in the longitudinal direction than in the transverse direction compared with samples without compatibilizer. Regarding puncture resistance, there is no difference between compatible and non-compatibilized samples. With respect to the impact of the fall dart, which is a measure of the impact strength of a film, there is a clear advantage of the compatibilized films (1.5% PI-*g*-MA obtained 75% of the gain and 6% PI-*g*-MA obtained 55% of gain) compared with the non-compatibilized sample. This gain is visible in the greater resistance that the film has in relation to its breakage. 

## 4. Conclusions

This work transforms a mixture of PE/PA from residues of a multilayer packaging film into a monolayer film with PI-*g*-MA as a compatibilizer. The resulting films were extensively characterized in terms of their morphology, thermal behavior, and physical properties. FTIR analysis confirmed that the cyclic anhydride from MA reacted during processing. SEM analyses showed differences between the non-compatibilized sample 0% PI-*g*-MA with a fibrous structure and the compatibilized samples with a more homogeneous and rougher structure. The compatibilized films showed an increase in elongation at break and a small gain in terms of tensile strength. Films with 1.5% and 6% compatibilizer show better properties in the longitudinal direction and 75% gain and 55% gain in the impact of the fall dart, respectively, compared with samples without a compatibilizer. These facts are indicative of the compatibilizing effect of the PI-*g*-MA polymer. 

## Figures and Tables

**Figure 1 polymers-16-01079-f001:**
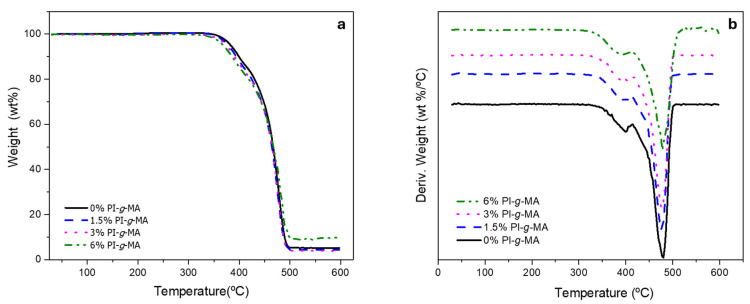
TGA thermograms of samples: (**a**) weight vs. temperature and (**b**) weight derivate vs. temperature.

**Figure 2 polymers-16-01079-f002:**
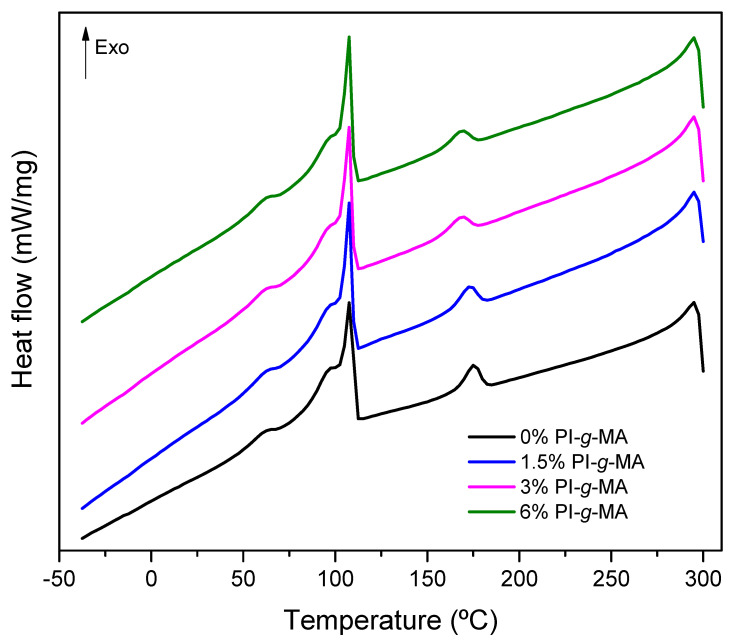
DSC thermogram of first cooling flux of films with different percentages of PI-*g*-MA.

**Figure 3 polymers-16-01079-f003:**
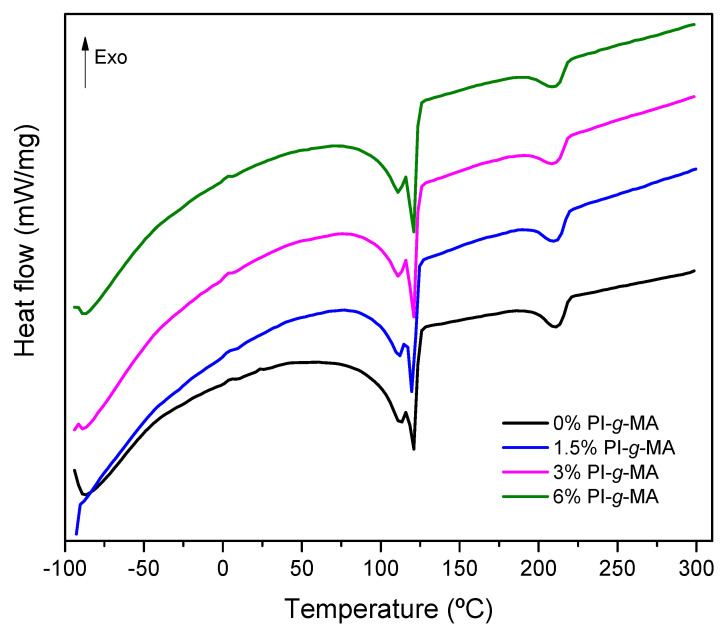
DSC thermogram of second heat flux of films with different percentages of PI-*g*-MA.

**Figure 4 polymers-16-01079-f004:**
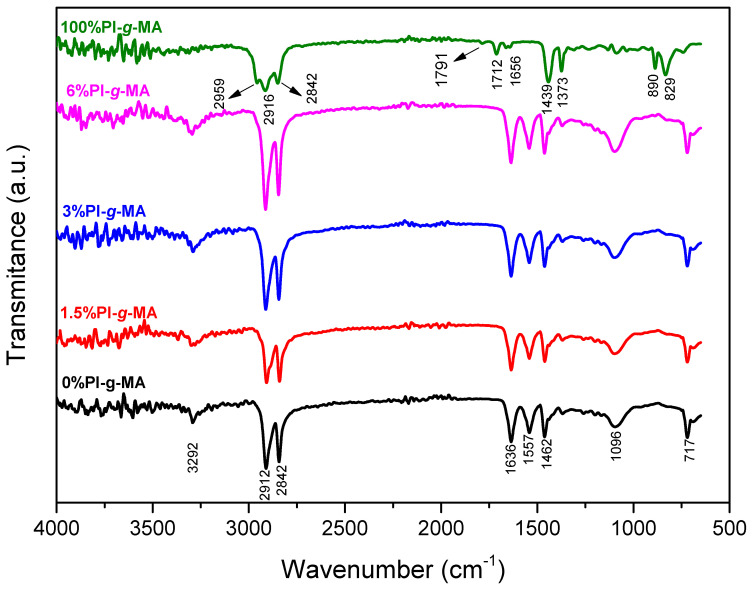
ATR-IR spectra of films of (PE/PA)rec with different percentages of PI-*g*-MA and only PI-*g*-MA.

**Figure 5 polymers-16-01079-f005:**
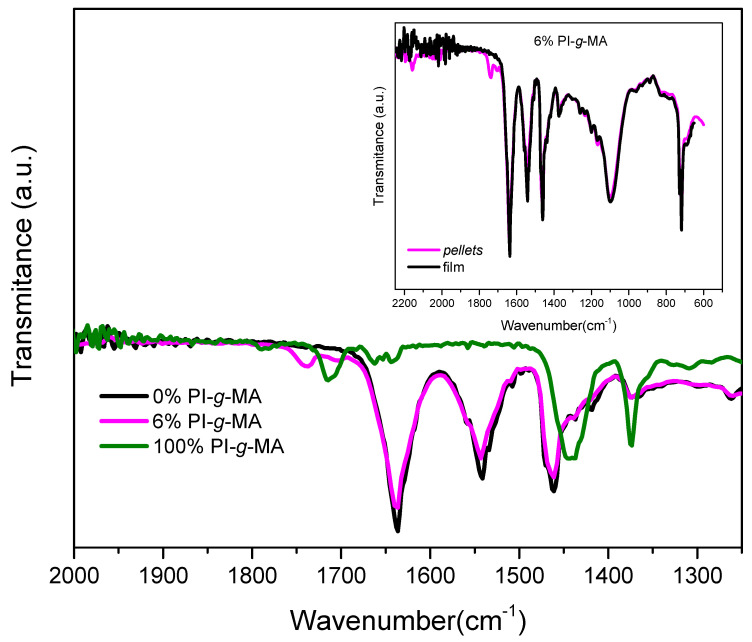
ATR-IR spectra of pellets of (PE/PA)rec with different percentages of PI-*g*-MA and insert ATR-IR spectra of the film (two processings) and pellets (one processing) of a sample with 6%PI-*g*-MA.

**Figure 6 polymers-16-01079-f006:**
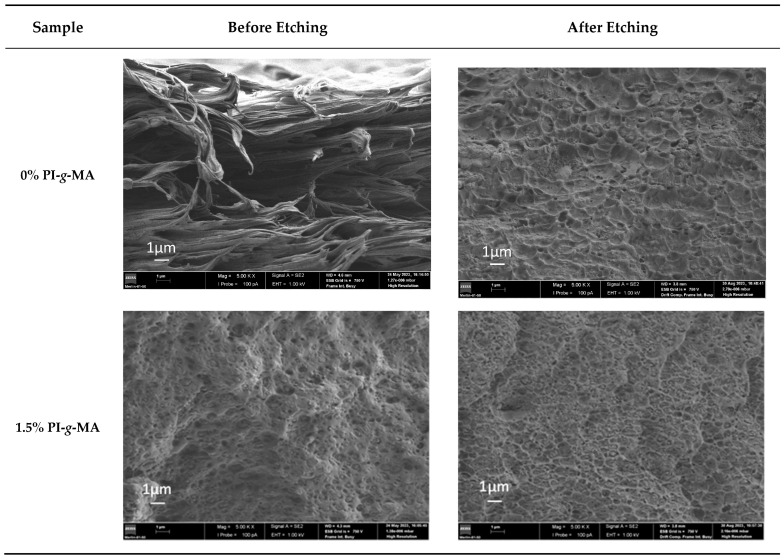
SEM micrographs of the cryofracture surface of the (PE/PA)rec with different percentages of PI-*g*-MA.

**Figure 7 polymers-16-01079-f007:**
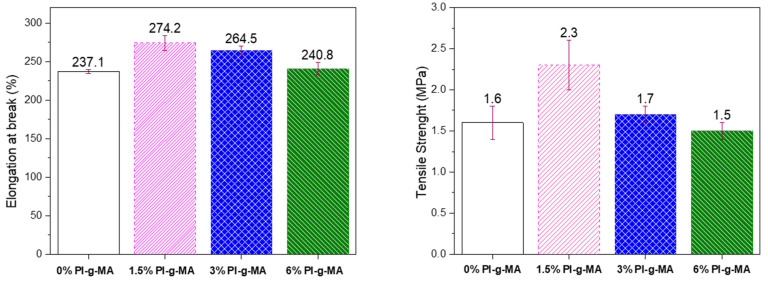
Elongation at break and tensile strength of the films.

**Table 1 polymers-16-01079-t001:** Temperature profile.

Temperature Profile (°C)							
Zones	Zone1	Zone 2	Zone 3	Zone 4	Zone 5	Zone 6	Head
Set-Point	100	180	195	220	220	220	220

**Table 2 polymers-16-01079-t002:** Extrusion conditions.

Spindle Speed (rpm)	Feeder (%)	Cutting Speed (%)
290	5	8

**Table 3 polymers-16-01079-t003:** Film production parameters.

Temperature Profile (°C)						Extrusion Parameters	
Zones	Zone 1	Zone 2	Zone 3	Zone A	Zone B	Pull	Spindle Speed (%)
Set-Point	190	220	220	220	220	2.7	27

**Table 4 polymers-16-01079-t004:** Result obtained using TGA analysis.

Sample	1st Loss		2nd Loss		Residue (%)
	T_onset_ (°C)	%	T_onset_ (°C)	%	
0% PI-*g*-MA	372.2	15.0	447.3	80.0	5.0
1.5% PI-*g*-MA	343.7	16.0	453.1	80.0	4.0
3% PI-*g*-MA	369.3	17.7	454.8	78.6	3.7
6% PI-*g*-MA	365.5	17.8	469.1	72.5	9.7

**Table 5 polymers-16-01079-t005:** DSC data of second heat flux of the films with different percentages of PI-*g*-MA.

Sample	Polyethylene				Polyamide			
	T_m_ (°C) *	∆H_m_ (J/g)	Tc (°C)	∆Hc (J/g)	T_m_ (°C)	∆H_m_ (J/g)	Tc (°C)	∆Hc (J/g)
0% PI-*g*-MA	120 (113)	85.6	108	55.2	211	62.8	175	48.0
1.5% PI-*g*-MA	120 (112)	85.6	107	71.1	211	59.8	172	41.0
3% PI-*g*-MA	120 (112)	83.3	107	61.7	210	57.7	168	35.1
6% PI-*g*-MA	120 (112)	88.3	107	64.7	211	61.1	168	32.2

* in parenthesis, the values for the observed shoulders.

**Table 6 polymers-16-01079-t006:** Results of tear resistance, puncture resistance, and resistance to the impact of the fall of the dart.

Sample	Tear Resistance (g)		Puncture Resistance		Impact of the Fall Dart (g)
	Longitudinal	Transverse	Energy (J)	Load (N)	
0% PI-*g*-MA	576 ± 42.1	828 ± 18.8	0.037 ± 0.0	15.40 ± 1.07	440
1.5% PI-*g*-MA	608 ± 46.2	719 ± 12.5	0.036 ± 0.0	15.27 ± 2.0	770
6% PI-*g*-MA	748 ± 38.8	816 ± 10.3	0.039 ± 0.0	14.61 ± 0.87	680

## Data Availability

No special data support this article. Appendix A is available.

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
