# Peer review of "Recycling Polyethylene/Polyamide Multilayer Films with Poly(isoprene-g-Maleic Anhydride) Compatibilizer"

_polymers, 2024, doi:10.3390/polym16081079_

Round 1
Reviewer 1 Report (Previous Reviewer 2)
Comments and Suggestions for Authors
In the revised version, this referee did not see much significant improvement to recommend its acceptance. For examples, after numerous comments to improve the original manuscript, the revised version still showed 8 SEM micrographs with invisible scale bars (they presented the SEM graphs on a table - Table 5); Figs. 2 and Fig. 3 still have miss-spelled words in figure captions (DCS), etc… Those are just some examples; there are many more; this referee does not want to spend excessive time to act as a proof-reader.
The entire manuscript is more like a test report, with little science aspects. For example, Conclusion states: “Films with 1,5 % and 6% of compatibilizer shows better properties in the longitudinal direction and 75% of gain and 55% of gain in impact of the fall dart, respectively compared with sample without compatibilizer These facts are indicative of the compatibilizing effect of the PI-g-MA polymer.” There is little science merit to justify such “properties”, which made the manuscript more like a company’s laboratory test report. Even though they attempted to use SEM morphology results to “support” it [claiming: SEM analyses showed differences between the non-compatibilized sample 0% PI-g-MA with a fibrous structure and the compatibilized samples with a more homogeneous and rougher structure.], as a matter of fact, this referee does not see any meaningful science background that “Films with 1,5 % and 6% of compatibilizer shows better properties”, etc…
In the discussion texts, there are many cases where they cited irrelevant/incoherent refs. for "supporting" their interpretations that are actually scientifically wrong. These scientific flaws are difficult to be remedied, even with revisions.
Comments on the Quality of English Languagewriting needs polishing.
Author Response
Reviewer 1
In the revised version, this referee did not see much significant improvement to recommend its acceptance. For examples, after numerous comments to improve the original manuscript, the revised version still showed 8 SEM micrographs with invisible scale bars (they presented the SEM graphs on a table - Table 5); Figs. 2 and Fig. 3 still have miss-spelled words in figure captions (DCS), etc… Those are just some examples; there are many more; this referee does not want to spend excessive time to act as a proof-reader.
Answer: It is sad that you think so, because as you can see (by the colors), the text has been revised. The proposed examples have exaggerated the situation. The scale of the SEM is shown and the misspelled word is DCS instead of DSC. I am sorry for the time you have spent.
The entire manuscript is more like a test report, with little science aspects. For example, Conclusion states: “Films with 1,5 % and 6% of compatibilizer shows better properties in the longitudinal direction and 75% of gain and 55% of gain in impact of the fall dart, respectively compared with sample without compatibilizer These facts are indicative of the compatibilizing effect of the PI-g-MA polymer.” There is little science merit to justify such “properties”, which made the manuscript more like a company’s laboratory test report. Even though they attempted to use SEM morphology results to “support” it [claiming: SEM analyses showed differences between the non-compatibilized sample 0% PI-g-MA with a fibrous structure and the compatibilized samples with a more homogeneous and rougher structure.], as a matter of fact, this referee does not see any meaningful science background that “Films with 1,5 % and 6% of compatibilizer shows better properties”, etc…
In the discussion texts, there are many cases where they cited irrelevant/incoherent refs. for "supporting" their interpretations that are actually scientifically wrong. These scientific flaws are difficult to be remedied, even with revisions.
Answer: We disagree with the catastrophic vision of the referee. On some points we have difficulty in explaining all the facts, but an attempt has been made to justify with results presented in the literature.
Reviewer 2 Report (New Reviewer)
Comments and Suggestions for Authors
This study investigated the compatibilization performance of PI-g-MA in polyethylene/polyamide blends. This paper was well written and can be published after a few minor revisions are made:
1. Please include other nonreactive compatibilization work in the introduction. (e.g. Proc Natl Acad Sci USA.120(34):e2301352120)
2. Figure 1: 6% PI-g-MA sample has more residues left, why?
3. Table 4 could include the crystallinity of PA and PE.
4. Table 6: Why 3% sample is not included?
5. In conclusion, which sample (compatibilizer loading) has the best overall performance?
Author Response
reviewer 2
This study investigated the compatibilization performance of PI-g-MA in polyethylene/polyamide blends. This paper was well written and can be published after a few minor revisions are made:
- Please include other nonreactive compatibilization work in the introduction. (e.g. Proc Natl Acad Sci USA.120(34):e2301352120)
A: Done
- Figure 1: 6% PI-g-MA sample has more residues left, why?
A: As indicated in the experimental part, it was necessary to use silica to process the (PE/PA)rec sample with PI-g-MA. Up to 3 % compatibilizer, 3 % silica was used, but at 6 % compatibilizer it was necessary to add a little more (+1.25 %), so the 6 % sample has a larger amount of residue due to the larger amount of silica in its processing.
- Table 4 could include the crystallinity of PA and PE.
A: The crystallinity of PE and PA is not given in the table, as we work with industrial waste consisting of different kinds of PE and PA and whose composition of PE and PA is variable and lies between 70-80% PE.
- Table 6: Why 3% sample is not included?
A: Aware by the referee’s advice we review the samples used for the properties listed in table 6. The tests were carried out in industrial environment and a large piece of film was used. For the film with 3% compatibilizer we found that the films break in the same point which corresponds to an inherent fragility of the material (generally with half the thickness of the rest of the film). This situation is due to a defect in the blow extruder on that day. In other films the defect was not detectable or is less frequent, so a defect-free piece could be cut. This is the main reason why the 3% film has poorer properties. For the mechanical tests the samples are smaller, and we can cut test specimens without defects. We have decided to remove the industrial results for this sample in table 6.
- In conclusion, which sample (compatibilizer loading) has the best overall performance?
A: The sample that has the best properties in all the tests performed is the sample with 1.5% compatibilizer, which means that we do not need to add a large amount of PI-g-MA to obtain better results, so we can conclude that 1.5% compatibilizer has the best overall performance.
Reviewer 3 Report (New Reviewer)
Comments and Suggestions for Authors
After review the manuscript, there are several points that need to be improved or corrected, before the manuscript can be accepted, following there are the detailed comments:
-Please define abbreviations first time is written. TG, DSC, LDPE, etc
- Verbs of actions in section 2, must be in past, not in present.
- I consider that explanation about how an extruder worsk must be deleted (lines 98 to 104), just indicate that a double screw extruder was used and conditions are reported in tables 1 and 2.
- Line 199 must be "films have"
-In experimental section indicate that only one heating cycle was applied forDSC, however in results indicate that there are two heating cycles, please clarify.
-In line 144, please change accumulations by scans.
- for SEM characterization, please indicate the accelaration voltage.
- for tensile testing, the dimension of specimens were according any ASTM or ISO procedures?
- for dart impact test, and tear test, how many replies were carried out??
- I recommend that table 1 (SI) must be inserted in main manuscript
- For TGA results discussion, there is not a discusion about the thermal stability, i mean is higher or lower comparing 0% PI-g_MA and materilas with 1,5, 3, 6% of modifier.
- In DSC results discussion,
-If Tc of PA decreases isgood or bad?? how this affects the material properties?
- Usually first heating cycle is used to delete the thermal history of polymers, so in this case i recommend to delete the figure 2 in main manuscript and move the figure S1 of supplementary file to main manuscript, with this supplementary file must be deleted considering my recommendation to move table S1 to main manuscript.
- For temperatures reported in Table 4, I recommend to delete decimals after point, due they consider negligible.
- For FTIR spectra (Figure 4) i recommend to process the spectrum with smoooting to reduces noise/signal ratio due there is lot of noise specially at high wavenumber, also please short the wavenumber scale to 4000 cm-1, due there are not signals at higher wavenumber.
- In line 257 indicate the absence of carboxylic anhydride signal, at which wavenumber must appears this signal? please indicate it.
- Figure 5 caption must indicate which lines corresponds to first and sceond processing, as it is indicated in lines 260 to 265. this in the aim to make easy to reader the identification of samples
- I recommend to change the table 5 to figure.
-In line 297 indicate that adding PI-g-MA there is an increase, however according with figure 6, this increase is only for sample 1.5 and then decrease for tensile strength even being lower than sample without compatibilizing
-In line 332 indicate that there is a clear advantage of the compatibilized films comparing with non-compatibilized, but why present this advantage? i mean what happends to material that shows this improvement.
- Please follow Instructions for authors of Polymers journal to report references.
Author Response
Reviewer 3
After review the manuscript, there are several points that need to be improved or corrected, before the manuscript can be accepted, following there are the detailed comments:
-Please define abbreviations first time is written. TG, DSC, LDPE, etc
A: Thank you for the observations. All abbreviations were defined the first time it appears.
- Verbs of actions in section 2, must be in past, not in present.
A: Thank you for the observations. The verbs of actions have been placed in the past.
- I consider that explanation about how an extruder worsk must be deleted (lines 98 to 104), just indicate that a double screw extruder was used and conditions are reported in tables 1 and 2.
A: Thank you for the observations. The authors thought it would be better for the reader to describe the process, but as indicated it becomes simpler for the reader to just indicate that a double screw extruder and the conditions used.
- Line 199 must be "films have"
A: Done
-In experimental section indicate that only one heating cycle was applied for DSC, however in results indicate that there are two heating cycles, please clarify.
A: Thank you for the observations. As you rightly indicated, the description of the second warm-up was missing from the experimental section. The section has been revised and this information has been placed in the text.
“Also, thermal behavior was evaluated by differential scanning calorimetry (DSC) made in a NETZSCH DSC 204 F1 Phoenix model (Netzsch, Germany). All samples were analyzed in an aluminium pan with an ordinarily closed aluminium lid. The samples were heating from room temperature to 300 °C, then cooled to -50 °C and followed a heating cycle to 300 °C. A heating/cooling/heating rate of 10 °C·min−1 was used. A dry nitrogen environment with a purge flow was applied.”
-In line 144, please change accumulations by scans.
A: Done
- for SEM characterization, please indicate the accelaration voltage.
A: Thank you for the observations. The acceleration voltage used is 1kV. This information has been indicated in the text for the reader's better understanding.
“To investigate the compatibilization the polymeric blend, the fracture surfaces were analyzed by scanning electron microscopy (SEM). The specimens were frozen in liquid nitrogen prior to fracture to diminish the risk of plastic deformation. The fracture surfaces were coated with gold and analyzed with 1kV of acceleration voltage in a Field Emission Scanning Electron Microscope (FESEM), ZEISS MERLIN Compact/ VPCompact, Gemini II.”
- for tensile testing, the dimension of specimens were according any ASTM or ISO procedures?
A: Yes, DIN EN ISO 527
- for dart impact test, and tear test, how many replies were carried out??
A: For dart impact test, and tear test we have made 7 replicas, and only the 5 bests were considered. These tests are carried out according to the standards ISO 7765-1, MET A and ISO 34 respectively.
- I recommend that table 1 (SI) must be inserted in main manuscript
A: Thank you for the observations. We have inserted the table 1 (SI) in main manuscript.
- For TGA results discussion, there is not a discusion about the thermal stability, i mean is higher or lower comparing 0% PI-g_MA and materilas with 1,5, 3, 6% of modifier.
A: The differences in thermal decomposition are so small between samples that no evident effect of compatibilizer is seen.
- In DSC results discussion,
-If Tc of PA decreases is good or bad?? how this affects the material properties?
A: The changes in the Tc of PA could indicate an interaction between PA and PE and improve the properties of PE. An important aspect is that we measure the effects in a specific situation, which in most cases does not correspond to the industrial environment. The cooling rate used is very different from industrial cooling rates, but these aspects are not well studied by academy, and in my opinion the most important ones are.
- Usually first heating cycle is used to delete the thermal history of polymers, so in this case i recommend to delete the figure 2 in main manuscript and move the figure S1 of supplementary file to main manuscript, with this supplementary file must be deleted considering my recommendation to move table S1 to main manuscript.
A: Thank you for the observations. As suggested by reviewer, we have inserted the second heating in the main manuscript and also table S1. In our opinion, the presence of the first heating is important if we want to understand the behavior of the material we hold in our hands, which results from processing. Processing can change the material properties due to the heating and cooling rates, which can trigger or prevent crystallization processes. During the first heating, the consequences of these processes become visible. The second heating, which erases this history, could tell me “how far” we can go with this particular mixture.
- For temperatures reported in Table 4, I recommend to delete decimals after point, due they consider negligible.
A: Thank you for the observations. We have deleted the decimals after point.
- For FTIR spectra (Figure 4) i recommend to process the spectrum with smoooting to reduces noise/signal ratio due there is lot of noise specially at high wavenumber, also please short the wavenumber scale to 4000 cm-1, due there are not signals at higher wavenumber.
A: Thank you for the observations. We have processed the spectrum with smoothing to reduces noise/signal.
- In line 257 indicate the absence of carboxylic anhydride signal, at which wavenumber must appears this signal? please indicate it.
A: The values are indicated on the text.
- Figure 5 caption must indicate which lines corresponds to first and sceond processing, as it is indicated in lines 260 to 265. this in the aim to make easy to reader the identification of samples
A: Thank you for the observations. We have indicated in the legend the processing’s.
Figure 5. ATR-IR spectra of pellets of (PE/PA)rec with different percentages of PI-g-MA and insert ATR-IR spectra of film (two processing’s) and pellets (one processing) of a sample with 6%PI-g-MA.
- I recommend to change the table 5 to figure.
A: Thank you for the observations. We have change table to figure.
-In line 297 indicate that adding PI-g-MA there is an increase, however according with figure 6, this increase is only for sample 1.5 and then decrease for tensile strength even being lower than sample without compatibilizing.
A: Thank you for the observations. The text has been revised in order to make it more explicit to the reader that the increases are in relation to non-compatible samples and that in relation to tensile strength only samples with 1.5% and 3% have small gains
The mechanical performance of the samples was evaluated by tensile-strain tests. The results are presented in Figure 7. The results show that, in general, when adding PI-g-MA there is an increase in elongation at break compared with the sample without the PI-g-MA. Also, in relation to tensile strength there is a small gain for the compatibilized samples with 1.5% and 3% of compatibilizer. These facts are indicative of the compatibilizing effect of the PI-g-MA polymer.
-In line 332 indicate that there is a clear advantage of the compatibilized films comparing with non-compatibilized, but why present this advantage? i mean what happends to material that shows this improvement.
A: Thank you for the observations. This gain is visible in the greater resistance that the film has in relation to its breakage.
- Please follow Instructions for authors of Polymers journal to report references.
A: We follow the rules of Polymers and also compared with articles already published.
Round 2
Reviewer 3 Report (New Reviewer)
Comments and Suggestions for Authors
After review the corrected version of manuscript, this shows a significant improve, due authors consider and correct most of the comments done.
Author Response
Thank you
This manuscript is a resubmission of an earlier submission. The following is a list of the peer review reports and author responses from that submission.
Round 1
Reviewer 1 Report
Comments and Suggestions for Authors
Serra et al. present a study on the utilization of commercial polyisoprene-graft-maleic anhydride as a compatibilizer in the recycling of PE/PA multilayer films. The mechanical recycling of such multilayer films is an economically viable and straightforward method for reusing waste plastic films.
The manuscript is suitable for publication in polymers; however, there are some issues that require attention before it can be published.
1. In Figure 5, visible bands corresponding to maleic anhydride (1703 cm-1, 1738 cm-1) are evident for (PA/PE)rec6%PI-g-MA pellets, but no signal is observed for the corresponding film. The ATR-IR spectrum of the samples is unclear. A critical question arises: how is the IR spectrum of the pellets characterized?
2. The results indicate that films with 1.5% and 6% PI-g-MA exhibit better properties in the longitudinal direction compared to the blank sample(Table 7). However, the performance of the sample with 3% PI-g-MA shows a decrement. It is imperative for the authors to elucidate the reasons behind this decrease in performance for the 3% PI-g-MA sample.
Comments on the Quality of English LanguageThe quality of English in this paper is well.
Reviewer 2 Report
Comments and Suggestions for Authors
Comments
This work is quite application-oriented, but with little science base. Writing needs dramatic improvement. Tech. flaws are listed as following.
1. Terminology or abbreviations for samples are quite confusing, such as: (PA/PE)rec6%PI-g-MA pellets, (PE/PA)recX%PI-g-MA, etc. Thus, such terms in Figure caption texts become confusing too. For example: Figure 5. ATR-IR spectra. of (PE/PA)rec6%PI-g-MA film and pellets.”, etc…….
2. Miss-spelling and grammar syntax errors are quite extensive. Even the caption of y-axis or Fig. 5 is wrong. Syntax and spelling errors such as: with different amount of PI-g-MA were measure using TGA. (line 200). What is “The (PE/PA) rec pellets….”; What is the phrase: “..has 100 (symbol) film thickness” (line 115)? What is “The recycled material (PE/PA) rec used in this study….” (line 89)?, etc.
3. What is meant by: “extrusion conditions to obtain pellets Tables 3 and 4” (line 114)? Syntax errors such as: “In the Tables 1 and 2 show the temperature profile and extrusion conditions to obtain the pellets.” (line 102). What is meant by: “….in order to get the films In Table 5 are shown the equipment parameters for the extruder.”? These phrases/sentences are incredibly amazing and confusing! Similar errors are extensive. List would be endless, and this referee cannot exhaust all. It appears writing is not properly or carefully guided.
4. What is “polyethylene vinyl alcohol”? Is it properly expressed? Syntax error again in: “In the case of food packaging multilayer films used to maintain food integrity and protect against degradation processes.” (line 28). The term of “ethylene vinyl acetate” (line 32) is a monomer; can it be used for “ low-temperature heat seal ability”, as authors stated?
5. Art work in Fig. 4 needs improvement. The four sub-graphs are jammed, with the legends on graphs or x/y-axes becoming too tiny to be visible.
6. In the literature, researchers do not indicate glass transitions by using “tg (°C)”, or melting temperature using tm1, tc1, etc.
7. Fig. 2 and Fig. 3 (DSC) can be merged into one.
8. Technically, which direction of y-axis is endo., or exo. in Fig. 2 and Fig. 3?
9. Not just those comprehensive editorial/writing discrepancies as commented above, technically, the assignments of IR bands in Fig. 4 are also erratic. For example, author discussed: “The band at 1460 cm-1 is identified to the C-C bonds [32].” By the way, citation of “104466-104466” is wrong, but this is not the main point. Throughout the entire texts of ref. 32, this referee has never been able to identify where texts of “The band at 1460 cm-1 is identified to the C-C bonds” were located? (Fig. 1 of ref. 32). That is, authors either cited a totally wrong ref.32, or fabricated the discussion statement.
10. Moreover, two questions to authors – regarding their discussion of “The band at 1460 cm-1 is identified to the C-C bonds [32],” are the C-C bonds able to produce IR absorption? and if yes, would it be at 1460 cm-1? Why would not author consult to fundamental textbooks of IR? Instead of ref. 32?
11. If IR assignment on the simplest C-C is in obvious error, it can be expected that there are many other similar scientific errors in the other IR band interpretations too. Authors stated: “The adsorption bands at and 1542 cm-1 are associated to amide II (-NH-CO- group) and amide I (C=O stretch), respectively.” [line 256]. First of all, the statement of “bands at and 1542 cm-1 are…..” is in error. Secondly, something is missing between “at” and “and”. If just ignoring the syntax errors, however, scientifically, the 1542 cm-1 band is associated with which one? - with the amide II (NH-CO- group) or amide I (C=O stretch)? Readers are totally lost.
12. Or, alternatively, did author mean to indicate that the 1542 cm-1 band is associated with the amide II (NH-CO- group)? NH-CO- is a group, and not a single bond. IR band is associated with a bond vibration, not a group. Authors pls. clarify clearly which bond of the “amide-II (-NH-CO- group)” is to be discussed?
13. Furthermore, how can the C=O stretch vibration be at 1542 cm-1? –Authors should check college textbooks; C=O stretching has never been reported at 1542 cm-1. That is to say, the 1542 cm-1 band is related to something else.
14. Contrary to the authors’ discussion texts “The band at 1095 cm-1 indicated skeletal aliphatic C-H rocking vibration [35,36]”, the absorption band of C-H rocking is never at 1095 cm-1! Authors even cited refs.-35,36 that have never made such interpretation at all! For example, Fig. 3 of Ref. 35 showed IR data, but never mentioned: “The band at 1095 cm-1 indicated skeletal aliphatic C-H rocking vibration”, etc. Citing overtly wrong or totally irrelevant refs for “justifying” their erroneous assignment of IR bands is one of the major tech/scientific discrepancies in this writing. The entire writing/discussion of this manuscript is in incredible technical errors!
Comments on the Quality of English LanguageEnglish in writing has extensive problems, with details shown in comments to authors.
Reviewer 3 Report
Comments and Suggestions for Authors
In this manuscript, the authors reported the application of a commercially available PI-g-MA polymer as compatibilizer in the blends of PE and PA. However, the results showed by the authors didn’t support what the authors’ claim. The PI-g-MA polymer may not introduce any potential compatibilization in the blends, leading to similar mechanical properties of blends compared to that of no compatibilizer. Thus, I recommend rejection of the paper without further consideration.
1. In DSC figures, the authors should note in the y axis if the endo is up or down.
2. In the DSC sections, the authors discussed about the addition of compatibilizers would change thermal properties of the blends. However, based on the figures and table 6, the glass transition temperatures, meting temperatures, and crystallization temperatures didn’t show statistical significant difference to support the authors’ conclusion. Instead, the authors should compare the melting enthalpy and/or crystallization enthalpy to reflect any potential difference. However, based on the figures, the addition of compatibilizers didn’t affect the thermal properties of blends at all.
3. FTIR of compatibilizer should also be included for comparison.
4. From the SEM images, it is hard to conclude the addition of compatibilizer resulted in less phase segregation. All images look similar in terms of macrophase separation.
5. From mechanical properties, it is also very difficult to conclude the addition of compatibilizer lead to better mixing in the blends, which is also reflected by the results in previous sections. Theoretically, the compatibilized blends should significantly enhance the elongation at break and toughness of the blends. However, with increasing addition of compatibilizer, the results should a completely different trend. Thus, it is very hard to conclude that this compatibilizer is useful in the system.